# Impact of Cell Design Parameters on Mechanical Properties of 3D-Printed Cores for Carbon Epoxy Sandwich Composites

**DOI:** 10.3390/polym17010002

**Published:** 2024-12-24

**Authors:** Mustafa Aslan, Kutay Çava, Altuğ Uşun, Onur Güler

**Affiliations:** 1Metallurgical and Materials Engineering, Karadeniz Technical University, 61080 Trabzon, Turkey; cavakutay@ktu.edu.tr; 2Medical Device Design and Production Application and Research Center, Karadeniz Technical University, 61080 Trabzon, Turkey; altug@ktu.edu.tr; 3Mechanical Engineering, Karadeniz Technical University, 61080 Trabzon, Turkey; 4HPET Engineering, 61081 Trabzon, Turkey; 5Advanced Engineering Materials Research Group, Karadeniz Technical University, 61080 Trabzon, Turkey

**Keywords:** sandwich composite, additively manufacture, cell design, compression flexural, cellular materials, auxetic design

## Abstract

The introduction of 3D printing technology has broadened manufacturing possibilities, allowing the production of complex cellular geometries, including auxetic and curved plane structures, beyond the standard honeycomb patterns in sandwich composite materials. In this study, the effects of cell design parameters, such as cell geometry (honeycomb and auxetic) and cell size (cell thickness and width), are examined on acrylonitrile butadiene styrene (ABS) core materials produced using fusion deposition modeling (FDM). They are produced as a result of the epoxy bonding of carbon epoxy prepreg composite materials to the surfaces of core materials. Increasing the wall thickness from 0.6 mm to 1 mm doubled the elastic modulus of the re-entrant structures (5 GPa to 10 GPa) and improved compressive strength by 50–60% for both geometries. In contrast, increasing cell size from 6 mm to 10 mm significantly reduced compressive strength by 80% (from 2.5–2.8 MPa to 0.5–0.6 MPa) and elastic modulus by 70–78% (from 9–10 GPa to 2–3 GPa). Flexural testing showed that the re-entrant cores, with a maximum load capacity of 148 N, exhibited more uniform deformation, while the honeycomb cores achieved a higher load capacity of 273 N but were prone to localized failures. These findings emphasize the directional anisotropy and specific advantages of auxetic and honeycomb designs, offering valuable insights for lightweight, high-strength structural applications.

## 1. Introduction

The study of cellular geometries, particularly honeycomb and re-entrant (auxetic) patterns, caught the interest of advanced material investigators due to their distinct mechanical properties and prospective uses.

These cellular patterns, which are frequently used in sandwich panel construction, are critical in industries ranging from aerospace to automotive engineering, where material efficiency and performance are essential. Honeycomb structures, known for their excellent strength-to-weight ratio and energy absorption properties, have long been utilized in various applications, including aircraft fuselage panels, automotive crash structures, and lightweight building components [1]. Their inherent geometric design provides superior rigidity and compressive strength, making them indispensable in such high-performance engineering fields [2,3,4].

However, the ability to study complex geometries such as re-entrant structures has opened up new possibilities in cellular design with the introduction of 3D printing technology. Re-entrant (auxetic) geometries, defined by their negative Poisson’s ratio, behave in a reverse manner under applied loads, expanding laterally when stretched and contracting when compressed [5]. This uncommon characteristic yields materials with improved fracture resistance, indentation hardness, and energy absorption [6]. Liu [7] highlighted the potential application of auxetic materials in aircraft, but their practical implementation remains constrained by manufacturing challenges.

The rise of 3D printing technology has considerably reduced these limitations, opening up new possibilities for the fabrication of both honeycomb and re-entrant structures with an enhanced capacity for design flexibility. The influence of wall thickness on the mechanical properties of 3D-printed cores has been demonstrated in numerous studies [8,9]. For instance, research has shown that ABS cores with thicker walls produced via fused deposition modeling (FDM) exhibit higher tensile strength. However, these studies have primarily focused on individual parameters, without exploring their interplay in complex cellular systems.

Comparative investigations into the mechanical performance of cellular structures reveal a gap in the understanding of their behavior under diverse loading scenarios. Yap et al. [10] investigated the compressive strength of various cell shapes, demonstrating that geometry has a major influence on polylactic acid (PLA) material properties, but they limited their scope to honeycomb structures. Similarly, Chun Lu and colleagues [11] investigated a range of grid patterns, concluding that Quadri-Grid designs demonstrate superior mechanical properties. However, a direct comparison between honeycomb and re-entrant geometries in sandwich composites under identical conditions remains relatively underexplored.

Gohar et al. [12] highlighted the potential of composite face sheets paired with 3D-printed ABS cores to enhance flexural resistance and interfacial bond strength, highlighting FDM’s capacity, but their analysis did not extend to auxetic core materials. Li and Wang [13] investigated the integration of honeycomb and re-entrant geometries in sandwich composites through numerical analysis. Nevertheless, their study did not address the influence of density variations or the interaction of geometric features with material properties.

Li and Wang [13] conducted a study integrating 3D printing and numerical analysis with design sandwich composites with diverse bending behaviors, incorporating both honeycomb and re-entrant geometries. Their research provides a comprehensive understanding of how cellular design influences the overall performance of sandwich panels. Similarly, Wang et al. [14] examined the mechanical properties of core materials with various unit cell shapes produced via 3D printing. Their findings indicated that triangular unit cell structures exhibited the lowest compressive strength, which they attributed to their higher susceptibility to plastic deformation from limited intersecting cell wall surfaces. Furthermore, the compressive strength of honeycomb-patterned cores was found to be significantly impacted by variations in density, which were driven by cell wall thickness and dimensions.

Existing studies on the mechanical performance of 3D-printed cellular structures have made significant progress in understanding how geometric parameters influence material behavior. Research has demonstrated that cell size, wall thickness, and overall topology critically affect properties such as compressive strength, stiffness, and energy absorption [13,14,15,16,17]. Traditional honeycomb structures, known for their high stiffness-to-weight ratios, are effective for specific applications but are limited in their energy absorption capabilities [14]. Despite these advancements, several gaps in literature remain. The studies often focus on specific topologies or materials without systematically comparing honeycomb and auxetic geometries under consistent parameters. On the other hand, auxetic (re-entrant) geometries, with their negative Poisson’s ratio, offer enhanced fracture resistance and impact absorption, highlighting their potential as alternatives to conventional designs [13,15].

Collectively, these studies provide a comprehensive understanding of how cell design parameters influence the mechanical properties of 3D-printed core materials in sandwich composites. However, in this study, the researchers investigate how 3D printing’s capability to create diverse geometries and flexible cell sizes, particularly through honeycomb and auxetic patterns, serves as an innovative alternative to enhance the core and overall performance of carbon epoxy-based sandwich composites, marking a pioneering comparison in the field.

## 2. Materials and Methods

### 2.1. Design of Sandwich Core Cells

The core materials were designed using Solidworks, a computer-assigned design (CAD) program. The designs were converted to .stl (standard triangle language) format using the Solidworks program so that the slicer required to operate the 3D printers could be transferred to the programs. Typical honeycomb and re-entrant cell designs were determined as references in selecting the core material designs to compare them with the most common commercial auxetic and cellular geometries. The geometry used in the cell design of the honeycomb and re-entrant patterns is given in Figure 1.

The geometric parameters of the honeycomb and the re-entrant cell design are given in Figure 1. The typical cell parameters were determined to be a cell wall angle of 30 degrees and cell lengths (h) of 6 and 12 mm for the re-entrant and honeycomb cell structures, respectively. The cell geometry of the core materials with all the auxetic and cellular patterns was designed to have a cell length of 6 mm and a wall thickness of 0.6 mm. Honeycomb and re-entrant structures with wall thicknesses of 0.6, 0.8, and 1 mm and cell lengths of 6, 8, and 10 mm were determined to investigate the effect of cell length and wall thickness on the compression behavior of the ABS sandwich core materials with the relative density and experimental density of the re-entrant and honeycomb cores, as shown in Table 1. The relative density for each cell geometry can be calculated for the honeycomb and re-entrant patterns.
(1)$$\frac{p*}{p_{s}}=\frac{t/l(h/(l+2))}{2cos\theta (h/(l+sin\theta ))}$$

In Equation (1), *t* represents the wall thickness, *l* the unit cell wall length, and ℎ the height of the cell, with *θ* accounting for the angular geometry specific to the core design. It is important to note that the core structures of honeycomb and re-entrant types were maintained with equivalent unit cell wall lengths, specifically at dimensions of 6 mm, 8 mm, and 10 mm, ensuring a consistent basis for comparison.

Table 1 presents a detailed comparative analysis of these core structures, providing insights into the impact of varying geometric configurations. By altering the cell length (CL) and wall thickness (WT), we can observe the direct influence these fundamental parameters have on the physical properties of the cores. From the data presented, it is observed that re-entrant ABS cores exhibit higher density values relative to their honeycomb counterparts when compared at the same unit cell size and wall thickness. This suggests a greater material presence within the re-entrant structure, indicating a higher mass fraction of the solid material within the given volume.

The relative density, which is a ratio of the core’s density to the solid material’s density, and the density values themselves provide key insights into the structural composition of the cores. Table 1 reveals that the re-entrant ABS cores possess higher density values at each corresponding unit cell size and wall thickness compared to the honeycomb cores. This higher density in re-entrant cores suggests that they contain a greater amount of material within the same volume, which could imply a denser packing of the material and potentially affect the core’s mechanical properties.

### 2.2. Production of Core and Sandwich Materials

Different patterned core material sample designs converted to .stl format were transferred to the slicer program called Cura, which is open-source software that allows control of the print parameters in the 3D printer. The different three-dimensional patterned core material sample models were transferred to the Cura program for production on a 3D printer. The core materials were produced on the FDM (Fused Deposition Modeling)-type 3D printer (Ultimaker 3, BV Watermolenweg, The Netherlands), as shown in Figure 2. Each sample of the auxetic and cellular structures was printed using ABS (acrylonitrile butadiene styrene) raw material. However, the honeycomb and re-entrant structures had a 6 mm cell length and 0.6 mm wall thickness. To produce the carbon surface layers used in the manufacturing of sandwich composite materials, curing was performed under hot pressing, using autoclave-free prepreg material (VTP H300, SPM, Ankara, Turkey) that was suitable for achieving the desired fiber volume fraction (target thickness). For this purpose, carbon epoxy prepregs with a twill weave pattern, a 200 g/m^2^ fiber weight, and a 50% fiber volume fraction were utilized. Typically, carbon fibers used in such prepregs are continuous, running the full length of the material roll, and have diameters ranging from 5 to 7 μm. The epoxy resin has a glass transition temperature of 120 °C. The prepregs were layered in three plies and cured under 10 bar pressure, following the specified temperature and time curing schedule. Through this process, a fiber volume fraction of 55% was achieved in the produced composites. The surface layers were then cut from these composite plates to match the dimensions of the core material, using a circular saw with a diamond blade.

### 2.3. In-Plane Compression Testing of Sandwich Cores

Three specimens of each core material were used for compression testing. The compression test was carried out at a speed of 0.5 mm/min in the MTS universal testing machine with a 10 kN load cell according to the ASTM C364 standard, followed by stress–strain graphs. The elastic modulus is calculated using the initial linear region of the stress–strain curve, where strain is determined by dividing the crosshead displacement by the gauge length of the specimen. The tests were conducted in a quasi-static regime with a constant strain rate of 0.001 s^–1^. The deformation of the specimens at each level of compression loading conditions was recorded by a digital camera at 60 FPS. The honeycomb and re-entrant types (structure pattern and cell geometry) of core material were tested to represent the properties of printed ABS cores in the in-plane direction (Figure 3).

### 2.4. Compression Testing of Sandwich Composite

The compression testing of honeycomb and re-entrant sandwich panels was conducted following ASTM C365 to assess their mechanical properties in both the in-plane and out-of-plane directions. The specimens were prepared to standard dimensions, and testing was performed using a universal testing machine with flat compression plates. According to this standard, a compression test was conducted perpendicular to the surface of 50 × 50 ×12 mm sandwich specimens using an MTS universal testing machine, with a moving grip at a loading rate of 0.5 mm/min under a 10 kN load. Key metrics, including compressive strength, modulus of elasticity, and energy absorption, were analyzed.

### 2.5. Flexural Testing of Sandwich Composite

The flexural testing of the honeycomb and re-entrant sandwich panels was conducted following ASTM C393 to evaluate their bending properties. The specimens were prepared to standard dimensions, and testing was performed using an MTS universal testing machine equipped with a three-point bending fixture. In accordance with this standard, a flexural test was conducted by applying a load at the midpoint of the specimen’s span, with a loading rate of 1 mm/min under a 10 kN load capacity. Key parameters, including flexural peak load and deflection at peak load, were analyzed to assess the flexural performance and structural integrity of the sandwich composites.

### 2.6. Finite Element Analysis of Composite Tests

Finite element analysis (FEA) was performed in this study to validate a model and gain more insight into the deformation and failure mechanisms of the composites with the additively manufactured samples that were subjected to compression tests. In the modeling process the composite plates and cellular structures were meshed within HyperMesh, where the core and face plates were discretized using shell elements defined on the mid-surfaces of the solid model. To ensure detailed representation, all the components were meshed with first-order quadratic shell elements, with a mesh size of 0.5 mm. The thickness values of the shell elements were assigned theoretically using the SECTION_SHELL card. Initially, the shell elements were modeled with zero thickness.

This approach allows the same model to be utilized for any desired shell thickness in finite element analysis (FEA), providing flexibility and adaptability for various simulation scenarios. The meshed geometries can be seen in Figure 4. The model consists of the core, the top and bottom polymer layers, and the composite parts in the outermost section.

These analyses were carried out with LS-DYNA finite element software. In the modeling of the core, MAT_024_PIECEWISE_LINEAR_PLASTICITY was used. These parameters for ABS were taken as 1100 kg/m3 density, 2341 MPa elasticity modulus, 40.74 MPa yield strength, and 0.35 Poisson ratio, which are obtained from tensile tests. Material model MAT_054_ENHANCED_COMPOSITE_DAMAGE was used for the composite with the Chang-Chang damage criterion. In this material model, the material properties were taken from our previous study [17] for CFRP composites, which are shown in Table 2.

Contact between the core material and composites was modeled with AUTOMATIC_SURFACE_TO_SURFACE_TIEBREAK. This contact type was chosen to simulate the failure in the applied adhesive or core structure that removes the core from the composite plates. In the compression analysis, all the nodes of the composites were fixed in all directions while a constant linear displacement vector was given to the uppermost composite plate. FEA analyses were carried out with a linear implicit solution. The forces applied to all the composite plates were read from the results to ensure the static loading of both the compression and three-point bending tests. Nodal displacements were read from the uppermost plates. These force and displacement values were cross-plotted and divided, respectively, by the cellular structure area and length to obtain stress–strain diagrams.

## 3. Results and Discussion

### 3.1. Compression Behavior of Sandwich Cores

Due to structural anisotropy, the mechanical behavior of cellular materials is directionally dependent. The loading in the X direction (parallel to the cell walls) is used to assess the material’s direct strength, whereas the loading in the Y direction (perpendicular to the cell walls) is used to evaluate the structure’s capacity to disperse stress. The stress–strain analysis (Figure 5) demonstrates that the honeycomb structures exhibit ductile behavior with multiple failure points in the X direction, whereas the re-entrant structures demonstrate higher initial strength but less ductility, failing abruptly in the Y direction.

In the Y direction, the honeycomb structures experience successive buckling and failure of the cell walls, as reflected by the stress peaks and troughs, while the X direction demonstrates a more gradual stress decline. These observations align with FEM-based analyses [18], which support the anisotropic behavior of cellular structures and highlight the influence of cell orientation on load-bearing capabilities. This emphasizes the critical role of structural anisotropy in determining mechanical performance.

Total absorbed energy (TAE) for each sample was calculated using the formulation provided below, based on the data obtained from the force–displacement curves. In this equation, P(s) represents the force acting on the composite material as a function of displacement (s). The values are integrated from the origin (zero deformation) to the maximum deformation value achieved (b). The calculated values are presented in Table 3.

As shown in the Table 3, the out-of-plane samples demonstrated significantly higher energy absorption values in both the re-entrant and honeycomb configurations, as expected. Among the composite samples, the honeycomb in-plane configuration exhibited the lowest energy absorption, with a value of 38 J, whereas the honeycomb out-of-plane configuration displayed the highest energy absorption, reaching 289 J.
$$TAE=\int_{0}^{b}P\left(s\right)ds$$

Figure 6 shows that for both the honeycomb and re-entrant structures, as cell length increases from 6 mm to 10 mm, both compressive strength and elastic modulus decrease. This trend likely arises because larger cells are less resistant to compression and are more prone to buckling under load, making the material less effective at load bearing. The decrease in elastic modulus (stiffness) with longer cell lengths indicates increased flexibility in the material [19]. The reduction in both compressive strength and elastic modulus with increasing cell length is more pronounced in the re-entrant structure, possibly because the re-entrant geometry, characterized by a negative Poisson’s ratio, becomes significantly more compliant as cell length increases. At a 6 mm cell length, the re-entrant geometry exhibits slightly higher compressive strength than the honeycomb, suggesting that it may provide better strength characteristics at smaller cell sizes.

As shown in Figure 6a,b, increasing wall thickness enhances both compressive strength and elastic modulus in the honeycomb and re-entrant structures. Thicker walls provide more resistance to buckling and deformation, resulting in stronger and stiffer materials. The trend is consistent across both geometries; however, the re-entrant structure shows a slightly higher sensitivity to wall thickness, particularly in the elastic modulus, indicating that it benefits more in terms of stiffness as the wall thickness increases. For the honeycomb (bottom left), both compressive strength and elastic modulus increase as the wall thickness grows from 0.6 mm to 1 mm; this is likely due to the additional material support. Similarly, in the re-entrant structure (bottom right), these properties also increase with wall thickness, with a particularly notable rise in elastic modulus at 1 mm, underscoring the re-entrant design’s enhanced stiffness with thicker walls.

The mechanical performance of the honeycomb and re-entrant structures with varying cell sizes and wall thicknesses has been investigated, yielding similar results. Additionally, Ergene and Yalçın [20] emphasized that re-entrant structures, due to their negative Poisson’s ratio, are more sensitive to geometric changes, with wall thickness having a significant effect on their mechanical properties. Similarly, Széles et al. [21] examined doubly re-entrant auxetic structures, emphasizing the influence of geometric parameters on mechanical performance. The study confirms that reducing cell size and increasing wall thickness leads to higher compressive strength and stiffness, particularly in structures with negative Poisson’s ratios.

In summary, both the honeycomb and re-entrant core structures show that shorter cell lengths and greater wall thickness contribute positively to compressive strength and stiffness. However, the re-entrant structure’s mechanical properties appear to be more sensitive to these geometric changes; this is likely due to its unique deformation mechanics. These insights are crucial for designing cellular materials where specific mechanical properties are desired, such as in aerospace, automotive, and structural applications.

### 3.2. Flexural Behavior of Sandwich Structures

Figure 7 presents the flexural load and deflection behavior, and the deformation point of the carbon epoxy sandwich composite for a 3D-printed ABS core based on re-entrant and honeycomb geometries. Interpretation of the load:

Deflection graph: Point (a) on the graph corresponds to the initial linear region where both structures behave elastically. The steeper initial slope of the honeycomb curve indicates that it has a higher stiffness initially compared to the re-entrant core.

At points (b) and (b’), both structures start deviating from linear behavior, indicating the beginning of core material yielding. The honeycomb core (red curve) reaches a higher load before this deviation, suggesting that it can carry more load in the elastic region compared to the re-entrant core. The re-entrant core shows a sharp peak at (b’), followed by a significant drop, which suggests a sudden failure, potentially due to initial layer cracking or local buckling. Following the peak, the re-entrant core shows a more gradual decline in load-bearing capacity, with a series of smaller peaks and valleys (from (c) to (d)), suggesting that it can still carry some load even after initial failure. This is indicative of the auxetic behavior, where the structure can become denser and distribute stresses more evenly after initial failure. However, the honeycomb core shows a sharp decline after its peak load, indicating a more brittle failure mode. It does not exhibit the same degree of recovery as the re-entrant core, as can be seen in the smoother post-peak curve without secondary peaks.

The sequence of images labeled from (a) to (d) for the honeycomb and (a’) to (d’) for the re-entrant cores shows the progression of deformation under increasing load. Initially, both structures seem to maintain their shape with little visible deformation. As the load increases, the honeycomb structure starts to exhibit buckling, as seen in images (b) and (c), where the cells are beginning to collapse, and the panel exhibits noticeable bending. The re-entrant structure marked (b’) and (c’), appears to undergo a more uniform deformation without localized buckling, which is characteristic of materials with a negative Poisson’s ratio.

The flexural behavior of the sandwich panels demonstrated in Figure 7 suggests that the re-entrant core can show a higher area under a load deflection graph through its auxetic behavior, while the honeycomb core has a higher initial load-bearing capacity but a more brittle failure. The re-entrant core’s ability to carry a load after an initial failure could be beneficial in applications where a gradual failure mode is preferred, offering enhanced safety and energy absorption during impact or crash events

The three-point bending test results of the sandwich composites with honeycomb and re-entrant core patterns reveal distinct load-bearing behaviors, which are pivotal for their application in structural design. The observed higher maximum load capacity (273 N) in the honeycomb-patterned cores compared to their re-entrant counterparts (148 N) underscores the well-documented mechanical efficiency of honeycomb structures. This efficiency is attributed to the uniform distribution of stress and strain within the honeycomb geometry, which delays the onset of peak stress concentration and subsequent material failure. Conversely, the findings of H. Yazdani Sarvestan et al. [15] provide further context, demonstrating that auxetic re-entrant-patterned materials avoid local deformation and carry significant loads (up to 1200 N), though less than the maximum load determined by the octet-patterned core material. This observation suggests that while auxetic materials may not carry the highest loads initially, their post-failure performance could offer advantages in applications where a gradual failure mode is critical, such as impact resistance or energy absorption scenarios.

Interestingly, the post-failure analysis indicated that the honeycomb cores exhibited a reduction in load-bearing capacity after the initial breakage of the carbon epoxy surface layers, suggesting local deformation as a critical failure mode. This contrasts with the behavior of re-entrant-patterned composites, which, due to their auxetic properties, maintain load-bearing capabilities even after surface layer damage. Auxetic materials are known to exhibit a negative Poisson’s ratio, allowing them to become wider when stretched and contributing to their ability to absorb energy and maintain structural integrity post-damage [5]. In practical applications, the choice between honeycomb and re-entrant core patterns in sandwich composites should be informed by specific performance requirements. For structures that benefit from maximum load-bearing capacity without deformation, honeycomb cores are preferable. Conversely, in scenarios where a structure may need to maintain load bearing after initial damage, re-entrant auxetic patterns may offer superior performance due to their unique deformation characteristics.

### 3.3. Compression Testing of Sandwich Composites

In the out-of-plane direction, the re-entrant core reaches a peak stress more quickly, followed by a sharp decline, suggesting that failure occurs abruptly after the yield point (Figure 8a). However, the honeycomb structure, in contrast, shows a more gradual increase in peak stress and a slower decline after the peak, indicating a more ductile response with progressive failure, which may be due to the auxetic behavior allowing the structure to deform more before complete failure.

For in-plane compression, the honeycomb structure exhibits a lower initial stiffness than the re-entrant structure (Figure 8b). The stress increases to a higher peak before dropping off, which could be indicative of the walls buckling or the structure collapsing. On the other hand, the re-entrant structure’s stress initially overshoots that of the honeycomb before dropping below it and then surpassing it again. This behavior might suggest a more complex response to in-plane compression, possibly due to the auxetic nature of the material allowing the re-entrant structure to undergo a sequence of buckling and stabilization phases. Notably, the re-entrant structure seems to have a second peak in the in-plane direction, which is not observed in the out-of-plane compression. This second peak may represent a structural reconfiguration or a secondary load-bearing mechanism that is activated due to the in-plane loading conditions.

The deformation behavior suggested by the graphs (Figure 8) indicates that the honeycomb core deforms and fails in a more brittle manner, especially in the out-of-plane direction. This is consistent with the typical behavior of honeycomb structures, where once the cell walls buckle or fail, the material’s ability to carry further load drops sharply. The re-entrant core displays a more ductile deformation behavior, especially in the out-of-plane compression test. The material can take some load even after initial failure, which could be useful in applications where energy absorption is important, such as impact resistance.

The initial slope (modulus of elasticity) in both graphs shows that the re-entrant core is stiffer than the honeycomb core in both loading directions. The area under the curve, up to the failure point, represents the energy absorption capacity of the material. The re-entrant core seems to absorb more energy in the out-of-plane direction before failure, which can be advantageous in applications where energy dissipation is required. The honeycomb core’s higher peak stress in both loading directions implies that it can withstand higher loads before failure, but it also indicates a more sudden and less ductile failure.

The deformation behavior of the in-plane honeycomb and re-entrant models is illustrated in Figure 9, which showcases the finite element analysis (FEA) results of the honeycomb structure under a load and shows the re-entrant samples. The FEA results show a close resemblance to the deformation behavior observed in the actual tests. However, differences are evident due to factors such as the porosities and structural discontinuities inherently present in FDM-printed samples. These imperfections cause localized stress concentrations, leading to premature failure regions in some areas of the honeycomb structure, which are not accounted for in the idealized FEA model.

A notable distinction between the analysis and the real tests is observed from the homogeneity and mirrored deformation of the FEA analysis. In the actual printed samples, the deformation appears less uniform across the structure due to material inconsistencies and defects from the FDM process. These factors also contribute to deviations in the stress–strain curves beyond the yield point, with the analysis results showing higher stress values than the real tests. This occurs because of the idealized material assumptions in the FEA model, which neglects the effects of the microstructural imperfections that impact real-world performance. However, both the real and finite element models showed 45° slip planes in the honeycomb structures with a V-shaped deformation. Additionally, the composite plates performed as a rigid material that changed the deformation behavior of the samples, in both the real and FEA tests.

As expected, maximum von Mises stress concentrations are located at the nodal points within the cells, where moments peak due to the structure’s geometry. For the re-entrant models, a negative Poisson’s ratio behavior was observed in both the FEA and real tests, confirming the auxetic behavior characteristic of these structures. This property allows re-entrant models to contract when compressed, causing an accelerated densification phase behavior. This unique configuration enables them to reach a densification state more rapidly than their honeycomb counterparts

## 4. Conclusions

This study successfully demonstrated the fabrication of thermoplastic core structures with honeycomb and re-entrant auxetic patterns using an FDM-type 3D printer, integrated into sandwich composites with carbon epoxy prepreg layers. The mechanical performance of these cores was systematically evaluated under various load conditions, revealing key insights:The honeycomb cores excelled in out-of-plane energy absorption, achieving 288.79 J, outperforming the re-entrant cores by 251.68 J. During flexural testing, the honeycomb cores also supported a higher maximum load of 273 N, but their performance was marked by localized failures and sudden load drops, reflecting a tendency for brittle collapse.The re-entrant auxetic cores, in contrast, demonstrated superior resilience under in-plane loading, absorbing 107.80 J, nearly three times that of the honeycomb cores (37.51 J). Although their maximum load capacity in flexural testing was lower at 148 N, the re-entrant cores exhibited more uniform and controlled deformation, underscoring their ability to distribute stress more evenly.Increasing wall thickness from 0.6 mm to 1 mm doubled the elastic modulus of re-entrant structures (5 GPa to 10 GPa) and improved compressive strength by 50–60% for both geometries. Moreover, the smaller cells (6 mm) yielded much higher compressive strength (2.5–2.8 MPa) and elastic modulus (9–10 GPa) compared to the larger cells (10 mm), which experienced reductions of up to 80% in compressive strength and 70–78% in elastic modulus.

## Figures and Tables

**Figure 1 polymers-17-00002-f001:**
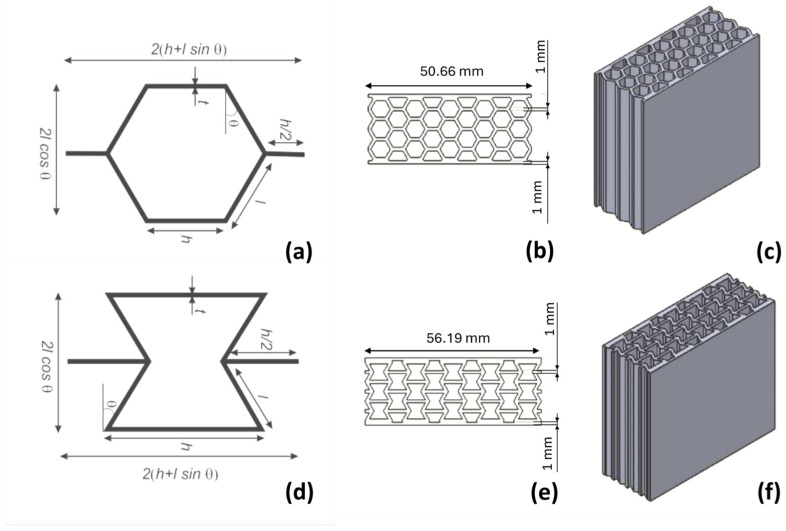
Typical honeycomb geometry; unit cell (**a**), in-plane (**b**), out-of-plane (**c**) direction and typical re-entrant core geometry; unit cell (**d**), in-plane (**e**), out-of-plane (**f**) direction.

**Figure 2 polymers-17-00002-f002:**
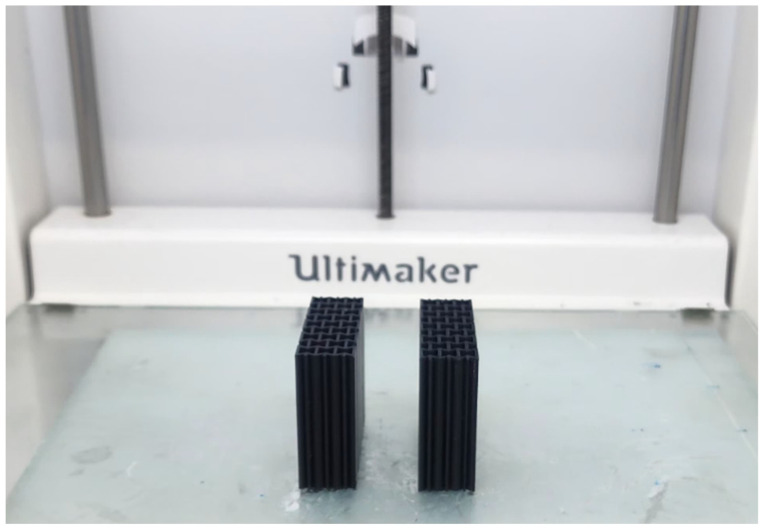
Printed sandwich core structures in X direction.

**Figure 3 polymers-17-00002-f003:**
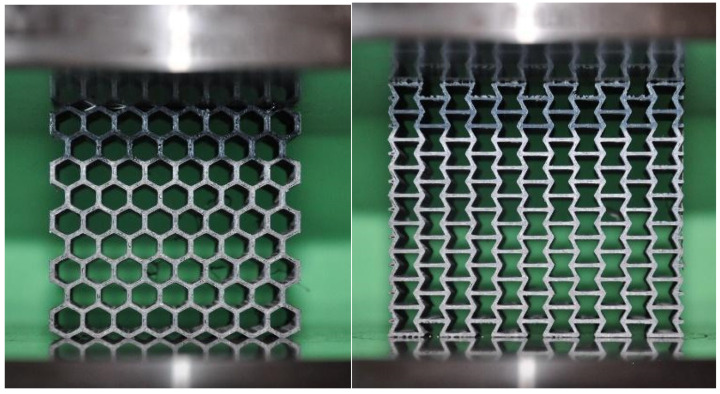
In-plane compression test on universal testing machine.

**Figure 4 polymers-17-00002-f004:**
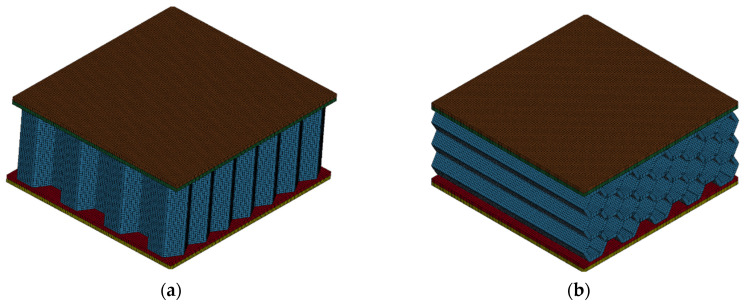
Meshed elements of compression test: (**a**) out-of-plane honeycomb, (**b**) in-plane honeycomb, (**c**) out-of-plane re-entrant, and (**d**) in-plane re-entrant.

**Figure 5 polymers-17-00002-f005:**
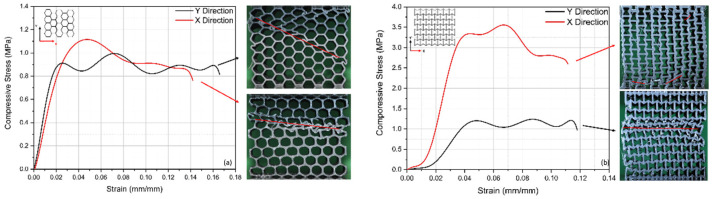
Stress–strain graphs of ABS core material: (**a**) honeycomb structure, (**b**) re-entrant structure in both in-plane (X and Y) directions.

**Figure 6 polymers-17-00002-f006:**
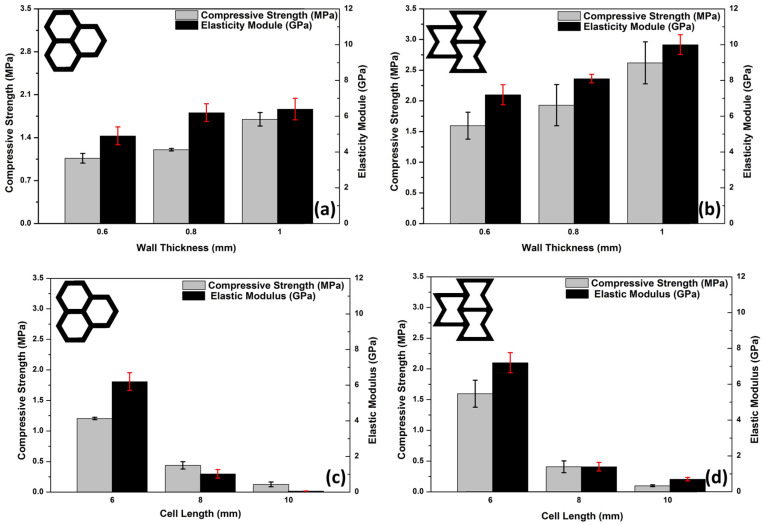
Compression strength and elastic modulus of 3D-printed ABS cores with different cell walls for honeycomb (**a**) and re-entrant (**b**) and wall thicknesses for honeycomb (**c**) and re-entrant (**d**) geometries.

**Figure 7 polymers-17-00002-f007:**
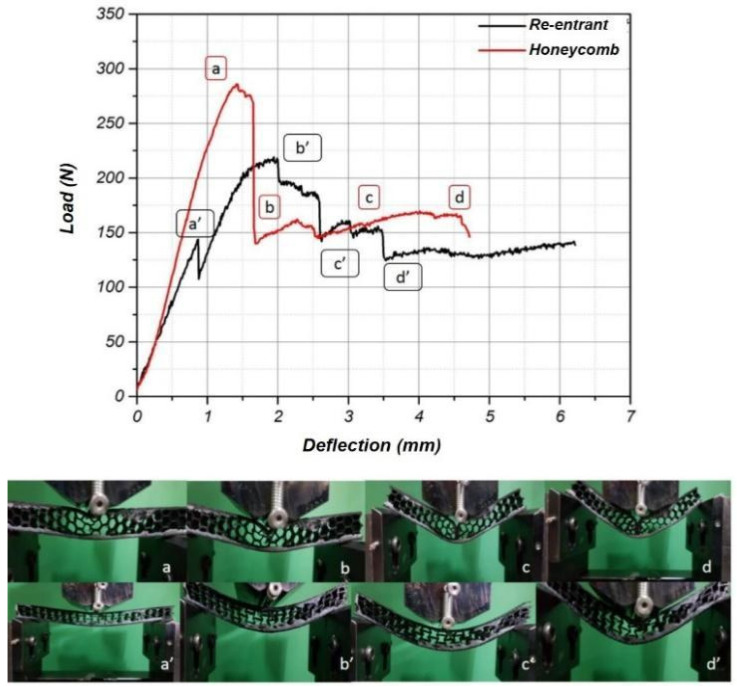
The flexural load and deflection behavior and sequential images of deformation points of carbon epoxy sandwich composite for re-entrant and honeycomb geometries.

**Figure 8 polymers-17-00002-f008:**
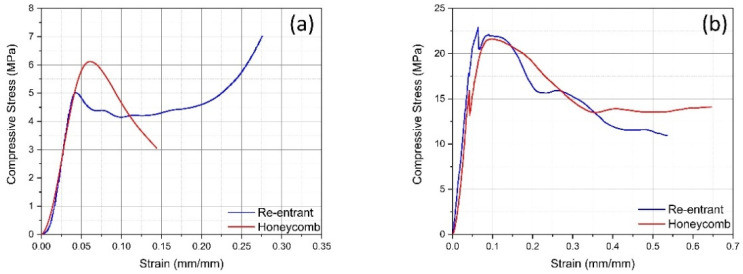
Stress–strain graphs of 3D-printed sandwich materials with honeycomb and re-entrant cores in different load directions: (**a**) in-plane, (**b**) out-of-plane.

**Figure 9 polymers-17-00002-f009:**
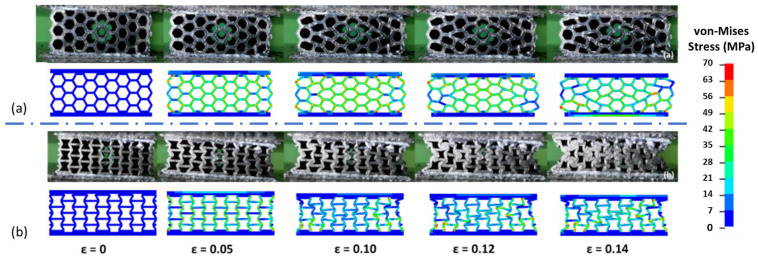
Compression of sandwich panels and finite element results: (**a**) honeycomb and (**b**) re-entrant.

**Table 1 polymers-17-00002-t001:** Relative density and density values of honeycomb and re-entrant ABS core structures.

Cell Length (CL) and Wall Thickness (WT) (mm)	Relative Density		Density	
	Honeycomb (*p**/p_s_)	Re-Entrant (*p**/p_s_)	Honeycomb (gr/mm^3^)	Re-Entrant (gr/mm^3^)
6 & 1	0.38	0.51	295	403
6 & 0.8	0.30	0.41	282	332
6 & 0.6	0.23	0.30	272	291
8 & 0.6	0.17	0.23	160	192
10 & 0.6	0.13	0.12	129	121

**Table 2 polymers-17-00002-t002:** FEA properties for composite material.

Property (Symbol)	Composite Material (Orthotropic)	ABS Polymer (Isotropic)
Longitudinal Stiffness (E1)	55.92 GPa	2.341 GPa
Transverse Stiffness (E2)	54.40 GPa	2.341 GPa
Shear Modulus (G12)	4.199 GPa	-
Poisson Ratio (v21)	0.043	0.35
Longitudinal Tensile Strength (Xt)	910.1 MPa	-
Longitudinal Compressive Strength (Xc)	710.2 MPa	-
Transverse Tensile Strength (Yt)	772.2 MPa	-
Transverse Compressive Strength (Yc)	703.2 MPa	-
Shear Strength (Sc)	131.0 MPa	-
Longitudinal Tensile Failure Strain (DFAILT)	0.0164	-
Compressive Tensile Failure Strain (DFAILC)	−0.013	-
Transverse Tensile Failure Strain (DFAILM)	0.014	-
Shear Failure Strain (DFAILS)	0.03	-

**Table 3 polymers-17-00002-t003:** Total absorbed energy values for sandwich composites.

Composite Sandwich	Loading Direction	Total Absorbed Energy (J)
Honeycomb	In-plane	37.51
	Out-of-plane	288.79
Re-entrant	In-plane	107.80
	Out-of-plane	251.68

## Data Availability

The original contributions presented in this study are included in the article. Further inquiries can be directed to the corresponding authors.

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
