# Peer review of "Impact of Cell Design Parameters on Mechanical Properties of 3D-Printed Cores for Carbon Epoxy Sandwich Composites"

_polymers, 2024, doi:10.3390/polym17010002_

Round 1
Reviewer 1 Report
Comments and Suggestions for Authors
In this study, thermoplastic core structures with honeycomb and reentrant auxiliary patterns were successfully fabricated using an FDM-type 3D printer, and their mechanical properties were evaluated under different load directions, providing insights into the mechanical performance of the two types of core structures. However, the innovation of this study is not good, the experimental volume is insufficient, and the following issues exist:
1. The introduction of the article is poorly structured and lacks clear logic; merely narrating other scholars' research is not conducive to writing an introduction. The literature review in the introduction should point out the shortcomings of existing studies to highlight the necessity and innovation of this research, but the authors fall short in this aspect.
2. In section 3.1 of the manuscript, the analysis of the reasons for the decrease in compressive strength and elastic modulus with increasing cell length lacks theoretical support; it is recommended to cite relevant literature to enhance persuasiveness. Additionally, it is not recommended to use vague terms to compare and analyze the differences between different structures.
3. The authors mentioned the research of these two scholars (Ergene and Széles). After reviewing the relevant literature, I request the authors to elaborate on the innovative points of this study and how they differ from the work of these scholars, as there is some overlap in the innovations, which makes the originality of this study confusing to me.
4. Regarding the analysis in section 3.3 of the manuscript, it is suggested that the authors include an analysis combining finite elements with macroscopic mechanical properties. Additionally, the appropriate introduction of energy calculations would further enhance the completeness.
5. In summary, after a comprehensive analysis of the manuscript, the reviewer still considers the study work relatively simple and suggests adding the aforementioned research content.
Author Response
In this study, thermoplastic core structures with honeycomb and reentrant auxiliary patterns were successfully fabricated using an FDM-type 3D printer, and their mechanical properties were evaluated under different load directions, providing insights into the mechanical performance of the two types of core structures. However, the innovation of this study is not good, the experimental volume is insufficient, and the following issues exist:
- The introduction of the article is poorly structured and lacks clear logic; merely narrating other scholars' research is not conducive to writing an introduction. The literature review in the introduction should point out the shortcomings of existing studies to highlight the necessity and innovation of this research, but the authors fall short in this aspect.
Respond: Thank you for your valuable feedback. The introduction has been thoroughly revised to improve its structure and logical flow. It now highlights the relevance of existing studies, clearly identifies gaps in the literature, and underscores the necessity and innovation of this research. These changes aim to provide a stronger foundation for the study, and the revisions have been marked for clarity.
- In section 3.1 of the manuscript, the analysis of the reasons for the decrease in compressive strength and elastic modulus with increasing cell length lacks theoretical support; it is recommended to cite relevant literature to enhance persuasiveness. Additionally, it is not recommended to use vague terms to compare and analyze the differences between different structures.
Respond: This section is summarised without vague terms to anbealyse the differences in first two paragraphs as marked in the text between 242 - 254 at page 8. In otder to support the reason of the behavior , the releavent paper is added to text as well. :
Öztürk, M., Baran, T. & Tatlıer, M.S. Experimental and numerical investigation of conventional and stiffened re-entrant cell structures under compression. J Braz. Soc. Mech. Sci. Eng. 44, 593 (2022). https://doi.org/10.1007/s40430-022-03889-x
- The authors mentioned the research of these two scholars (Ergene and Széles). After reviewing the relevant literature, I request the authors to elaborate on the innovative points of this study and how they differ from the work of these scholars, as there is some overlap in the innovations, which makes the originality of this study confusing to me.
Respond: The both scholar recited based on their findings and differences from this study and given in the text as shown below.The text was corrected as follows “Ergene and Yalcın [21] emphasized that re-entrant structures, due to their negative Poisson's ratio, are more sensitive to geometric changes, with wall thickness having a significant effect on their mechanical properties. Similarly, Széles, et al. [22] examines doubly re-entrant auxetic structures, emphasizing the influence of geometric parameters on mechanical performance. The study confirms that reducing cell size and increasing wall thickness lead to higher compressive strength and stiffness, particularly in structures with negative Poisson's ratios.”
- Regarding the analysis in section 3.3 of the manuscript, it is suggested that the authors include an analysis combining finite elements with macroscopic mechanical properties. Additionally, the appropriate introduction of energy calculations would further enhance the completeness.
Respond: We apologize if we may not have fully understood the macroscopic mechanical properties mentioned in the comment. If interpreted correctly, the material properties have been expanded upon in Section 2.6. Additionally, the energy formulation has been incorporated into the manuscript in Section 3.3. We sincerely thank the reviewer for this insightful comment, as it has significantly enhanced the quality and depth of the section.
- In summary, after a comprehensive analysis of the manuscript, the reviewer still considers the study work relatively simple and suggests adding the aforementioned research content.
Respond: Thank you for your comprehensive review and constructive suggestions. We acknowledge your feedback regarding the simplicity of the study and have incorporated the recommended research content.
Reviewer 2 Report
Comments and Suggestions for Authors
Paper No.: polymers-3343177
Title: Impact of cell design parameters on mechanical properties of 3D printed cores for carbon epoxy sandwich composites
There are some raised issues that should be treated.
1- Please include some numerical key results in abstract. The same guilt for the conclusions.
2- In line 35 : “ … making them suitable for a wide range of applications”. Please mention some specific parts of these applications.
3- Explain what is PLA as it has been mentioned for the first time in line 54? The same guilt for HCSS.
4- In line 97: “In low-speed impact tests, … ”. Could you please define the tests speed range?
5- In Fig. 1: the annotation (dimensions) text is not well visible. Also put the subfigure numbers (a) and (b) on the images. Moreover there are 6 imaged so that they should be numbered from (a) to (f). And the honeycomb’s drawing need to be placed in the lower raw against the 3D model.
6- The word reentrant is sometimes written with “ –“ and other times without. Please be consistent.
7- In line 158: “Ultimaker 3 model printer”, the authors may be mean “Ultimaker 3D model printer”.
8- In the description part of the carbon epoxy prepregs (lines 164-171) describe the fiber geometry (length and diameter).
9- In Fig. 3. Please change “Universal Test Machine” to “Universal Testing Machine”.
10- In the FEM model a mesh size of 0.5 mm is relatively large for a wall thickness of 0.6 mm. Why not to make the mesh size relative to the cell geometry?
11- The images of the FE models in Fig 4 are in low resolution. Please replace the images with higher resolution images.
12- In Fig. 5.b. due to the delayed true contact compression load, the curves should be shifter to the left direction with a strain value of around 0.012 mm/mm to start from zero like the curves in Fig. 5.a.
13- Please indicate how the elastic modulus measured without extensometer?
Author Response
Comment 1 Please include some numerical key results in the abstract. The same guilt for the conclusions.
Respond: Thank you for your valuable comment. Both the abstract and conclusion have been revised to include numerical key results for better clarity and specificity.
2. In line 35 : “ … making them suitable for a wide range of applications”. Please mention some specific parts of these applications.
Respond: Thank you for pointing this out. The text has been updated in the line between 35 to 40 to specify particular applications of honeycomb structures, including their use in aircraft fuselage panels, automotive crash structures, and lightweight building components, to provide more clarity and context.
3. Explain what is PLA as it has been mentioned for the first time in line 54? The same guilt for HCSS.
Respond: Thank you for your observation. The text has been updated to include the full name of PLA (Polylactic Acid) upon its first mention in line 61 for clarity. Similarly, the abbreviation HCSS has been fully deleted and explained with another reference in the following text.
4. In line 97: “In low-speed impact tests, … ”. Could you please define the test speed range?
Respond: Thank you for the comment. The reference to "low-speed impact tests" has been removed, and the explanation has been rephrased to provide a clearer and more detailed description of the testing conditions for better understanding.
5. In Fig. 1: the annotation (dimensions) text is not well visible. Also, put the subfigure numbers (a) and (b) on the images. Moreover, there are 6 imaged so that they should be numbered from (a) to (f). And the honeycomb’s drawing need to be placed in the lower raw against the 3D model.
Respond: Thank you for your insightful comment. The annotations in Fig. 1 have been revised to enhance visibility, and the subfigures have been numbered from (a) to (f) as suggested. Additionally, the honeycomb’s drawing has been repositioned
6. The word reentrant is sometimes written with “ –“ and other times without. Please be consistent.
Respond : Thank you for pointing out this inconsistency. The term has been standardized throughout the manuscript and is now consistently written as "re-entrant" .
7. In line 158: “Ultimaker 3 model printer”, the authors may be mean “Ultimaker 3D model printer”.
Respond: Thank you for your observation. The text has been clarified to correctly describe the equipment as "The core materials were produced using a Fused Deposition Modeling (FDM) type 3D printer (Ultimaker 3), as shown in Figure 2." This ensures the terminology is accurate and aligns with the context of the study.
8. In the description, part of the carbon epoxy prepregs (lines 164-171) describes the fiber geometry (length and diameter).
Respond: Thank you for your suggestion. The description of the carbon epoxy prepregs has been updated to include details about the fiber geometry. The text now specifies that the carbon fibers used in the prepregs are continuous, running the full length of the material roll, with diameters typically ranging from 5 to 7 micrometers. This addition provides greater clarity and technical precision.
9. In Fig. 3. Please change “Universal Test Machine” to “Universal Testing Machine”.
Respond: Thank you for your observation. The label in Fig. 3 has been corrected,
10. In the FEM model a mesh size of 0.5 mm is relatively large for a wall thickness of 0.6 mm. Why not to make the mesh size relative to the cell geometry?
Since the models were meshed using shell elements, the wall thickness values were defined hypothetically using theoretical equations. Consequently, the wall thickness value is not directly related to the cell geometry. Shell elements with zero thickness were initially modeled, allowing flexibility to assign any desired wall thickness in the SECTION_SHELL card in LS-DYNA. This approach enables the same model to be applied to various wall thickness values without altering the underlying geometry, ensuring adaptability and efficiency in the simulation process. We understand the vagueness of the manuscript and the corresponding information is added to the section.
11. The images of the FE models in Fig 4 are in low resolution. Please replace the images with higher resolution images.
Respond: - The images of the FE models in Fig 4 are in low resolution. Please replace the images with higher resolution images. The corresponding images were replaced with high resolution images. The corresponding images were replaced with high resolution images.
12. In Fig. 5.b. due to the delayed true contact compression load, the curves should be shifter to the left direction with a strain value of around 0.012 mm/mm to start from zero like the curves in Fig. 5.a.
Respond: In Fig. 5.b. due to the delayed true contact compression load, the curves should be shifted to the left direction with a strain value of around 0.012 mm/mm to start from zero like the curves in Fig. 5.a.
13. Please indicate how the elastic modulus is measured without an extensometer.
Respond: Elongation (strain) of the specimen was measured over a 50mm gauge length using an extensometer.
Round 2
Reviewer 1 Report
Comments and Suggestions for Authors
The reviewer considers that the revised manuscript is acceptable.
Author Response
Comment 1: The reviewer considers that the revised manuscript is acceptable.
Respond 1: Dear reviewer, on behalf of all the authors, we sincerely thank you for considering the revised manuscript acceptable. Your feedback and evaluation have been invaluable to us.

Reviewer 2 Report
Comments and Suggestions for Authors
In line 23: “with wall thicknesses of 0.6 0.8 and 1 mm”. Please use comma between 0.6 and 0.8.
The authors have not responded to the comments No. 12, in the previous revision:
In Fig. 5.b. due to the delayed true contact compression load, the curves should be shifted to the left direction with a strain value of around 0.012 mm/mm to start from zero like the curves in Fig. 5.a.
Author Response
Comment 1: In line 23: “with wall thicknesses of 0.6 0.8 and 1 mm”. Please use comma between 0.6 and 0.8.
Respond 1: Thank you for pointing out the issue on line 23. We have revised the text to include a comma between "0.6" and "0.8," as suggested. We appreciate your attention to detail.
Comment 2: The authors have not responded to the comments No. 12, in the previous revision, In Fig. 5.b. due to the delayed true contact compression load, the curves should be shifted to the left direction with a strain value of around 0.012 mm/mm to start from zero like the curves in Fig. 5.a.
Respond 2: Thank you for your observation and for pointing out that we had not addressed comment No. 12 in the previous revision. We sincerely apologize for the oversight. We have revised Fig. 5.b. as per your suggestion. The curves in Fig. 5.b. have now been shifted to the left by a strain value of approximately 0.012 mm/mm to ensure they start from zero, consistent with the curves in Fig. 5.a. We appreciate your valuable feedback and attention to detail, which have significantly contributed to improving our work.